# Nutrient and Mineral Compositions of Wild Leafy Vegetables of the Karen and Lawa Communities in Thailand

**DOI:** 10.3390/foods9121748

**Published:** 2020-11-26

**Authors:** Kittiyut Punchay, Angkhana Inta, Pimonrat Tiansawat, Henrik Balslev, Prasit Wangpakapattanawong

**Affiliations:** 1Department of Biology, Faculty of Science, Chiang Mai University, Huay Kaew Road, Chiang Mai 50200, Thailand; punchay.botany@outlook.com (K.P.); aungkanainta@hotmail.com (A.I.); tiansawat@yahoo.co.th (P.T.); 2Ecoinformatics and Biodiversity Group, Department of Biology, Aarhus University, Building1540, NyMunkegade114-116, DK-8000 Aarhus C, Denmark; henrik.balslev@bios.au.dk; 3Environmental Science Research Center (ESRC), Faculty of Science, Chiang Mai University, Huay Kaew Road, Chiang Mai 50200, Thailand

**Keywords:** ethnobotany, food analysis, indigenous food, micronutrients, nutrition

## Abstract

Wild food plants are commonly used in the traditional diets of indigenous people in many parts of the world, including northern Thailand. The potential contribution of wild food plants to the nutrition of the Karen and Lawa communities remains poorly understood. Wild food plants, with a focus on leafy vegetables, were ranked by the Cultural Food Significance Index (CFSI) based on semi-structured interviews. Twelve wild plant species were highly mentioned and widely consumed. The importance of the wild vegetables was mainly related to taste, availability, and multifunctionality of the species. Their contents of proximate and minerals (P, K, Na, Ca, Mg, Fe, Mn, Zn, and Cu) were analyzed using standard methods. The proximate contents were comparable to most domesticated vegetables. The contents of Mg (104 mg/100 g FW), Fe (11 mg/100 g FW), and Zn (19 mg/100 g FW) in the wild leafy vegetables were high enough to cover the daily recommended dietary allowances of adults (19–50 years), whereas a few species showed Mn contents higher than the tolerable upper intake level ( > 11 mg/100 g edible part). The wild leafy vegetables, therefore, are good sources of minerals and we recommend their continued usage by indigenous people. Further research on these wild leafy vegetables’ contents of antioxidants, vitamins, heavy metals, anti-nutrient factors, and food safety is recommended.

## 1. Introduction

The consumption of wild plants persists in many communities, especially among indigenous peoples to whom wild food plants are part of their traditional food systems [1]. Indigenous people often experience food insecurity and malnutrition, but local communities often possess traditional knowledge that can help them to alleviate these problems through harvesting, hunting, and gathering of wild plants [2,3]. Of the 380,000 different vascular plants known globally [4,5], only nine species represent two thirds (66%) of all the crop production. These nine species are maize (*Zea mays*), rice (*Oryza sativa*), wheat (*Triticum aestivum*), potato (*Solanum tuberosum*), soybean (*Glycine max*), oil palm (*Elaeis guineensis*), cassava (*Manihot esculenta*), sugar cane (*Saccharum officinarum*), and sugar beet (*Beta vulgaris*) [2]. In comparison, the importance of wild plants as a source of nutrients and minerals has been documented in many studies which show that they may cover required dietary intakes [6,7,8,9,10,11,12,13,14]. Moreover, wild plants not only have important roles in diets, but some of them also provide important health benefits with documented biological and pharmacological effects [10,15,16,17]. Therefore, renewed attention to wild plants, documenting their value, may increase food and health security in traditional communities.

Thailand has a high diversity of ethnicities, especially in the highlands in the north. The Karen are the largest ethnic group inhabiting the mountainous areas of northern Thailand [18]. The Lawa, in contrast, are one of the smallest ethnic groups, and they live scattered over northern Thailand [19]. The rotational swidden rice-based agricultural systems of the Karen and the Lawa have maintained many useful plants in microenvironments between their agricultural fields [20], and, in addition, they use a large number of forest plants for food, construction, fuel, medicine, animal food, clothes, and ritual purposes [21].

Many ethnobotanical studies from northern Thailand [20,21,22,23,24,25,26,27,28,29,30,31,32,33,34,35,36,37,38] describe a rich traditional knowledge which includes over 1700 plant species [39]. However, the research has not pinpointed the plant species that provide the daily dietary intake of these indigenous people. Still, the ethnic minorities in northern Thailand face food insecurity and malnutrition [40,41,42,43]. Promoting the consumption of traditional diets could improve nutrition because a number of wild food species, in particular ones that are used as vegetables, have been reported to have high nutritional qualities that surpass conventional vegetables [10,14,44,45]. However, the information on the nutritional composition of wild leafy vegetable species in Thailand is limited and incomplete. Knowledge of nutrition- or health benefits may improve the perception of and attitude toward wild food plants [46]. The nutrient composition of wild food plants may provide informative data that will be advantageous to the health- and food security not only for indigenous people but also the rest of the population.

Malnutrition is one of the challenges of public health when it comes to sustainable development goals (SDGs) in many developing countries [47]. Malnutrition increases the risk of chronic diseases, stunting, and, eventually, nutritional disorder. Furthermore, the deficiency of minerals, i.e., calcium and zinc, is recognized as a health problem worldwide based on food supply [48,49]. In Thailand, the number of under-nourished people between 2016 and 2018 was five million, whereas the prevalence of stunting children was ten percent, and five percent of wasting in the total population [3]. Most ethnic minorities experience mineral deficiency in their diets [40,41,43,50,51].

In this study, based on the food knowledge of the Karen and the Lawa, we determined the cultural food significance of wild leafy vegetable species for nutritional analysis. Nutrient compositions, which included proximate analyses (moisture, ash, protein, fat, fiber, and carbohydrate), and minerals composition (P, K, Ca, Mg, Fe, Na, Mn, Zn, and Cu), were evaluated. Nutrient composition data from our study should help in expanding nutrient databases and are beneficial for future research on food and human health.

## 2. Materials and Methods

### 2.1. Selection of Wild Leafy Vegetable Species

For this study, we used the Cultural Food Significance Index (CFSI) to evaluate the cultural significance of wild leafy vegetable species. The CFSI, developed by Pieroni in 2001 [52], is the sum of seven measures of food importance and is calculated as:CFSI = QI × AI × FUI × PUI × MFFI × TSAI × FMRI × 10^−2^(1)
where QI = quotation index; AI = availability index; FUI = frequency of utilization index; PUI = plant parts used index; MFFI = multi-functional food use index; TSAI = taste score appreciation index; and FMRI = food-medicinal role index [52,53]. To determine the QI, the number quotation of each useful species for particular use was counted. The interviewees were asked “How many of these vegetables can be found in the forest?” to determine the AI with the following categories: rare, medium, common, and very common. The FUI values were determined by asking about the frequency of eating these species. The PUI values express the multiple uses of a plant part. If multiple parts were collected, the value will be higher than the use of single plant part. The MFFI values were obtained by asking interviewees “How do you consume this vegetable?” High values were allocated to raw consumption or cooked alone species, and low values were obtained by using with other ingredients or condiments. The TSAI values were assessed by asking how much interviewees like these vegetables on the following scale: terrible, poor, fair, good, very good, and best. Some vegetable species had medicinal benefits in different roles. High values of FMRI were allocated with food as medicine, while low values were obtained from species that had no specification of therapeutic action, or where it was not recognized. All sub-indices were evaluated using Pieroni’s scales [52]. Wild leafy vegetable species with CFSI ≥ 300 in the Karen and Lawa communities were selected to analyze the proximate and mineral components.

### 2.2. Study Sites

Two Karen and two Lawa villages in Mae Chaem District, Chiang Mai, Thailand, were selected. This area is rich plant diversity, and local people traditionally gather wild food plants [29]. Wild vegetable species naturally grow in lower montane rainforest and fallows. The elevations range from 700 to 1100 m above sea level. The Cultural Food Significance Index (CFSI) was determined for the wild food plant species through semi-structured interviews with 48 Karen and 53 Lawa villagers (age range 20–85 years) in 2018 [29]. All procedures performed were in accordance with the International Guidelines for Human Research Protection and the International Society of Ethnobiology (ISE) Code of Ethics [54]. Informed consent was obtained from all participants. Ethical approval was obtained from the Chiang Mai University Research Ethic Committee (certificate of approval no. 019/61). Wild leafy vegetables were collected in the fields with the assistance of the informants and identified by the first author. All voucher specimens were deposited at the Department of Biology, Chiang Mai University Herbarium (CMUB) [29].

### 2.3. Preparation of Samples

For analysis, one kilogram of each plant species was sampled in the study villages from July to December 2019. The most-used plant part of each wild leafy vegetable species was selected for analysis. The homogenized samples were subsampled. After that, the fresh materials were rinsed and cleansed with deionized water. Only edible parts were kept and the excess deionized water was dried off at room temperature. Wild vegetable parts were packed in zip-lock bags. The analyses were performed at the Agricultural Technology Service Center, Chiang Mai University, on the day after collecting the vegetables. The scientific nomenclature of the plants used follow the World Flora Online [55]. The distribution and occurrence of all the plant species were retrieved from the Global Biodiversity Information Facility (GBIF, www.gbif.org). 

### 2.4. Chemical Analyses

Analyses of moisture content, ash, macrominerals, and trace minerals were carried out using standard methods of analysis of the Association of Official Agricultural Chemists (AOAC) [56] and reference methods for plant analysis [57]. Moisture content was measured following method 930.15 [56]. The plant samples were dried in a hot air oven at 135 °C for 2 h. The weights of the dried samples were measured and expressed as percentages of the moisture content. Ash content was determined following method 942.05 [56]. The samples were put into a crucible and placed in a temperature control furnace at 600 °C for two hours. The ash content was obtained by measuring the weight. Total protein was determined via nitrogen content by the combustion technique (LECO model FP-828) with a conversion factor of 6.25, following method 968.06 [56]. 

Crude fat extraction was determined using a Soxhlet apparatus and petroleum ether. Crude fat residues were measured after drying, following method 954.02 [56]. The crude fiber was estimated by the gravimetric method; the material was digested with sulfuric acid and sodium hydroxide solution, following method 962.09 [56]. Total carbohydrate was obtained by subtracting the value of water, crude protein, crude fiber, ash, and fat from 100, following method 991.43 [56,58]. Mineral elements analyses (phosphorus, potassium, sodium calcium, magnesium, iron, manganese, zinc, and copper) were performed using atomic absorption spectroscopy (Analytik Jena, model ZEEnit 700). About 1 g of dry ash of a sample was wetted with a small volume of deionized water and then brought into a solution using 2 mL of HCl. For determination of potassium, calcium, and magnesium, the final volume should be 100 mL, which is sufficiently above the detection limit for a 1.0-g sample. For determination of manganese, iron, copper, and zinc, the final volumes should be between 10 and 50 mL. Quantification of each element was performed using atomic absorption spectrometry and comparing the linear curve from the standard solution with the sample [57]. All analyses were carried out in triplicate.

### 2.5. Data Analysis

All sub-indices of CFSI were calculated by Microsoft Excel to indicate the cultural significance of the species. To determine the reason for the use of wild leafy vegetables, values of all sub-indices were given the same weight. The relationship between quotations and sub-indices were illustrated by Chord diagram in the R programming language. Each value from the nutrient analyses was obtained by calculating the average of three measurements, and data are presented as mean ± SD. The comparison of the means of minerals was determined by Tukey’s test and statistical significance was accepted at the *p* < 0.05 level. Nutrient contents are reported per 100 g of fresh weight compared to The Essential Guide to Nutrient Requirements by the National Academy of Sciences and the Dietary Reference Intake for Thais [59,60].

## 3. Results

### 3.1. Cultural Food Significance Index (CFSI)

According to the Cultural Food Significance Index, twelve wild food species had high CFSI values in both the Karen and the Lawa villages (Table 1). The folk names and traditional uses by the Karen and the Lawa of wild leafy vegetables are provided in Table 1. Most of the leafy vegetables originated in Asia, but a few species were more widely distributed, i.e., *Centella asiatica*, *Acmella uliginosa,* and *Musa acuminata*. The taste of the plant, the availability of plants, and multiple functionality were the main drivers of high significance, whereas the frequency of use, plant parts used, and medicinal roles were less important (Figure 1). The highest taste score was for *Oroxylum indicum* (15%), followed by *Spondias pinnata* (10%) and *Ficus auriculata* (9%). Most informants mentioned that the twelve wild leafy vegetables are very common and easy to collect and are available for a long collection period (Table 1). The multiple food uses of the twelve leafy vegetable species were cooked as soup (33%), eaten raw (31%), and boiled (10%). Most of the informants mentioned that the frequency of the leafy vegetable use was one time each month (36%), one time each week (21%), and more than one time each year but less than once per month (18%), respectively. The food medicinal role index values for the twelve wild leafy vegetables were smaller than their values for other sub-indices because the informants did not recognize the twelve wild vegetable species except *Centella asiatica*, which received a very high score.

### 3.2. Nutritional Properties

The proximate compositions of the twelve wild leafy vegetable species on a wet basis vary considerably (Table 2). Their moisture contents were in the range of 62.9–95.3 g/100 g fresh weight (FW). The lowest value was found in *Lygodium flexuosum* and the highest was for *Musa acuminata,* from which we sampled the fleshy inner pseudostem. The ash contents ranged from 0.79 to 4.28 g/100 g FW. The lowest ash content was obtained from *Oroxylum indicum* and *Senegalia rugata* and the highest was in *Spondias pinnata*. The protein contents ranged from 0.48 to 5.33 g/100 g FW. The highest protein content was observed in *Clerodendum glandulosum*, whereas the lowest protein content was observed in *Musa acuminata*. The fat contents ranged from 0.10 to 0.78 g/100 g FW, with the lowest in *M. acuminata* and the highest in *L. flexuosum.* The highest fiber content was found in the leaves of *Lygodium flexuosum* (11.82 g/100 g FW), while the lowest fiber content was observed in *Monochoria vaginalis* (0.79 g/100 g FW). The carbohydrate contents ranged from 1.54 to 17.94 g/100 g FW. The lowest carbohydrate content was found in *M. acuminata* and the highest was in *L. flexuosum*.

### 3.3. Minerals Content

The contents of macrominerals (P, K, Mg, Ca, and Na) and trace minerals (Fe, Cu, Zn, and Mn) in the selected wild leafy vegetable species vary widely among the species (Table 3). The potassium content was the highest in all the studied species and ranged from 221.7 to 1291.3 mg/100 g FW in *Senegalia rugata* and *Oenanthe javanica,* respectively. The calcium content was the second highest in leafy vegetables and ranged from 2.6 to 982.9 mg/100 g FW. The phosphorus content of the wild vegetables ranged from 9.3 to 77.7 mg/100 g FW. The highest phosphorus content was found in *Senegalia rugata*, while the lowest was observed in *Musa acuminata*. The magnesium content ranged from 6.0 to 104.4 mg/100 g FW. The highest magnesium content was obtained from *Centella asiatica*, while the lowest magnesium content was found in *Spondias pinnata*. The sodium content ranged from 0.4 to 21.2 mg/100 g FW, with the highest content obtained from *Centella asiatica* and the lowest content from *Musa acuminata*. The iron contents ranged from 0.3 to 11.1 mg/100 g. The highest iron content was found in *Acmella paniculata*, whereas the lowest content was observed in *Oroxylum indicum*. The manganese contents varied from 0.2 to 15.7 mg/100 g. Moreover, the copper content was the lowest in all the plant species. *Monochoria vaginalis* and *Musa acuminata* had copper contents below 0.01 mg/100 g fresh weight, while the highest copper content was found in *Oroxylum indicum* (0.41 mg/100 g FW).

## 4. Discussion

### 4.1. Cultural Food Significance Index

The wild leafy vegetable species had high CFSI values in both the Karen and the Lawa communities and these values were similar to those found for wild food plants in northwestern Tuscany, Italy [52]. They were, however, not as high as the values found for edible fruits and seeds in Bali, Indonesia, probably because the fruit species had multiple uses, including for medicine [53]. Time may influence the selection of wild food because fruits are more difficult to harvest than leafy vegetables. Besides, most wild leafy vegetables with high CFSI values had medicinal properties in both the Karen and Lawa communities (Table 1), which increased the sub-element of the CFSI scores [52,53,61,62]. Moreover, the availability of vegetables and the multifunctionality of food uses, i.e., fresh consumption, food addition, and condiment, are related to highly important species [52]. Using a compound index is a powerful tool to understand the cultural significance of resources in more detail, but informants have to deal with a large number of questions for every sub-element for each species known, and that is very time consuming [61]. The reasons for using wild leafy vegetables were mainly related to taste, availability, and multifunctionality, which is similar to trends in the use of wild vegetables in West Sumatra, Indonesia [46]. Taste is a sociocultural factor for wild food plant consumption which appears to be a prime motivation for continuing consumption of wild food plants [63], and this is related to changes in the use of wild food plants [64]. Moreover, loss of availability of familiar food taxa from the wild is one of the main reasons for a decrease in wild food plant consumption [64].

### 4.2. Proximate Composition

The moisture content of the 12 species studied fluctuated from 60 to 90%, which is usual in leafy vegetables [65,66,67,68]. The high moisture content reveals that leafy vegetables need appropriate preservation because they are likely to degrade by microbial activity [69]. The range of the ash contents was similar to that of wild vegetables in the western Himalayas [70] and India [71]. Besides, ash content indicates an excellent source of the mineral element in vegetables [13], because ash contains inorganic materials, including anions, cations, oxides, and salts [72]. The range of fiber contents was similar to wild food plants in India [71] but lower than wild food plants in Bangladesh [11]. The fiber contents of *Centella asiatica*, *Clerodendrum glandulosum*, and *Oenanthe javanica* (Table 2) were similar to fiber contents of domesticated leafy vegetables, i.e., Chinese cabbage (2.2 g/100 g FW), celery (1.6 g/100 g FW), and coriander (2.8 g/100 g FW) [73]. High fiber content has significant benefits for the digestive system, including regulation of intestinal function, glucose tolerance, and prevention of chronic diseases, such as colon cancer [74,75]. However, an excess of fiber causes a negative effect on mineral absorption of, for instance, iron [76]. The protein contents were comparable to other indigenous vegetables in India [68,70] and Thailand [12,67]. The protein contents of *C. gladulosum* and *Senegalia rugata* (Table 2) were higher than that of kale (*Brassica oleracea*), whose protein content was 4.28 g/100 g for the edible portion [73]. However, the protein contents were low when compared with the Dietary Reference Intake for Thai adults (0.9‒8.6% recommended per day) [60]. The content of fat in leafy vegetables were commonly reported to be from <1‒3 g/100 g [7,9,10,11,12,14]. The fat contents in the wild food plants in this study were all lower than 1%, similar to the edible flowers of Zingiberaceae in Thailand [67] and domesticated leafy vegetables, i.e., asparagus (0.1 g/100 g FW), lettuce (0.2 g/100 g FW), and spinach (0.4 g/100 g FW) [73]. Fats are involved in many vital processes in the body, such as the absorption of fat-soluble vitamins [77], which may be correlated with deficiency of fat, vitamin A, and vitamin E among children of the ethnic minorities in northern Thailand [42,43]. The average carbohydrate content was similar to values obtained for indigenous vegetables in southern Thailand [12]. Some wild vegetables, i.e., *Oroxylum indicum*, *Clerodendum glandulosum*, and *Centella asiatica*, have carbohydrate values that are higher than cultivated vegetables, such as cabbage (3.2 g/100 g FW) or carrot (4.9 g/100 g FW) [78], which contributed to the nutrition, food security, and health by substitute for resource-poor farmers in developing regions [10]. 

### 4.3. Mineral Content

Minerals are important in the physiological processes of the human body. Sodium and potassium function as electrolytes, which maintain ionic balance and establish the electric potential across cellular membranes [59,72]. According to the nutrient composition database of food made available by United States Department of Agriculture (USDA), many wild leafy vegetables show high contents of macrominerals and trace minerals when compared with domesticated leafy vegetables. The range of the sodium content in the 12 species studied here was quite low when compared with the recommended dietary intake for adults (0.02‒1.4% recommended per day) (Table 4) [59]. Low sodium contents in wild vegetables have been reported in previous studies of wild vegetables [70,72,79] and domesticated leafy vegetable, i.e., asparagus (2 mg/100 g), lettuce (5 mg/100 g), and Chinese cabbage (29 mg/100 g) [73]. In our data, the potassium contents were higher than what was found in indigenous vegetables in southern Thailand [12], and some species have moderate potassium levels (5‒27% of RDA) when compared with recommended dietary intakes. The calcium contents varied, but some species, such as *Spondias pinnata* and *Ficus auriculata*, had higher calcium levels in fresh weight than wild edible plants in Bangladesh [77,80]. Wild Asteraceae members in this study were high in calcium, similar to wild Asteraceae in traditional Mediterranean diets [14]. The contribution of leafy vegetables to calcium content showed the highest percentage in *Spondias pinnata* (98% of RDA). The calcium content of *S. pinnata* was six times higher than the calcium contents of Chinese cabbage (150 mg/100 g FW) and kale (152 mg/100 g FW) [73]. 

The consumption of wild plants, for instance, *Centella asiatica* (31–41% of RDA) and *Clerodendrum glandulosum* (21–27% of RDA) (Table 4), can help villagers to meet the recommended magnesium intake, which prevents gastrointestinal disorders, renal loss, endocrine and metabolic disorder, cutaneous loss, and redistribution of magnesium to the bones [59]. The iron contents in the wild vegetables were high and similar to leafy vegetables in Africa [13], India [70,72], and Bangladesh [77]. The contribution of leafy vegetables to improving iron intake is relatively high in terms of percentage of recommended dietary allowances, i.e., *Acmella paniculata* (61–138% of RDA), *A. uliginosa* (59–133% of RDA), *Monochoria vaginalis* (51–116% of RDA), and *Oenanthe javanica* (38–87% of RDA) (Table 4). According to the daily iron takes of Thai children, the range of iron intake is between 6.6 and 8.4 mg/day [49], whereas the recommended dietary allowance in children is 7–11 mg/day [59]. This causes Fe deficiency in Thai children in urban (32%) and rural (38%) areas [49]. However, it would take 50–100 g (fresh weight) of iron-rich species to reach the recommended volume, and the consumption of iron-rich species should be a food of choice for Thai children suffering from anemia caused by Fe deficiency. The copper levels of *Acmella* species were similar to the wild Asteraceae in Mediterranean countries, which were reported to be 0.01‒0.4 mg/100 g [14]. 

Interestingly, the manganese content was relatively high, surpassing the daily requirement for adults [60]. Manganese is a natural part of most foods [81]. Although manganese is essential to bone structure, metabolism, and function as a powerful antioxidant, a high level of manganese can cause harm to the nervous system, which, in turn, may cause insomnia, depression, or illusions and, eventually, progressive alterations in gait and balance and tremor and Parkinson-like symptoms [82]. According to the percentage of recommended dietary allowance of manganese, many wild leafy vegetables show a very high percentage, such as *Centella asiatica*, *Monochoria vaginalis*, and *Lygodium flexuosum*, while the manganese contents of domesticated leafy vegetables, i.e., asparagus, Chinese cabbage, celery, lettuce, spinach, and kale, show less than one milligram per 100 g fresh weight [73]. However, the consumption of manganese-rich species should be cautioned regarding tolerable upper intake level of manganese, but the risk of excess intake of manganese from food appears to be low [59]. Furthermore, the Department of Mineral Resources considered our study villages in Mae Chaem district as having manganese mineral deposits [83]. Zinc was found in high levels when compared with previous studies of wild plants [11,14,72,79]. A study of dietary intakes of 0.5–12-year-old Thai children indicated that more than 50% of urban and rural Thai children have a low intake of zinc—only 4 mg/day [49]—whereas the dietary reference intake of Thai children (0.5–12-year-old) ranged from 2.7 to 9.5 mg/day [60]. The consumption of a high percentage of zinc as per the RDA, i.e., *Centella asiatica*, *Oenanthe javanica*, and *Clerodendum glandulosum*, could prevent Thai children from the risks of chronic diseases, such as growth retardation, eye and skin lesions, and delayed sexual maturation and impotence [59].

## 5. Conclusions

Our study has revealed the importance of wild food plant species, particularly leafy vegetables, commonly used in traditional diets in northern Thailand. The importance of wild vegetables was mainly related to taste, availability, and multifunctionality of species, whereas the frequency of utilization, plant parts, and medicinal roles are less important. The proximate compositions are comparable with most domesticated leafy vegetables. It illustrates the potential of wild plants as sources of essential nutrients. These wild vegetables are rich in macrominerals and trace minerals when compared with recommended dietary allowances, including magnesium (41% of RDA), iron (138% of RDA), and zinc (242% of RDA). A few wild vegetable species showed high manganese contents that exceeded tolerable upper intake level ( > 11 mg/100 g edible parts), producing health risks. The data have been mostly quantitative with fresh leafy vegetables. This is important and underlines the need to pay attention to the quantity of wild plant selection and consumption. The determination of bioavailability of minerals, antioxidants, vitamins, heavy metals, and anti-nutrient factors from these wild plants is necessary to evaluate their safety. Pharmacological studies should also be conducted to verify some of the ethnomedicinal properties.

## Figures and Tables

**Figure 1 foods-09-01748-f001:**
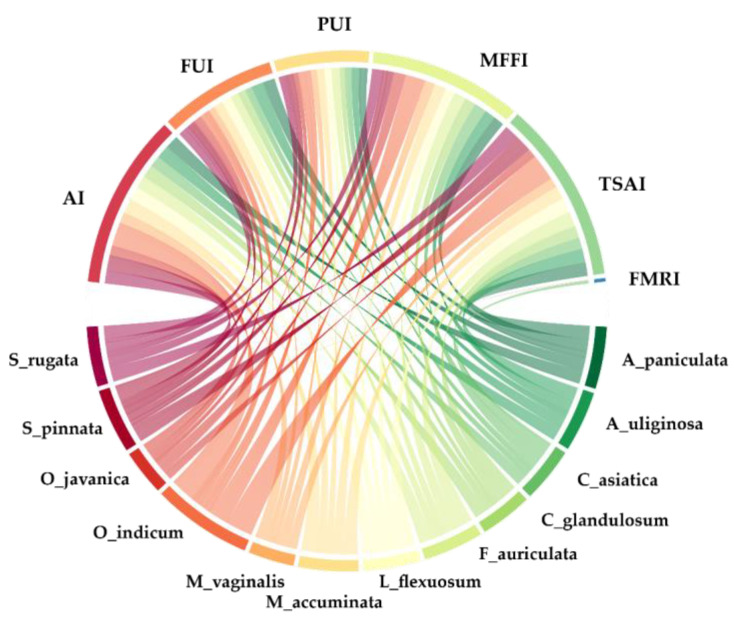
The relationship between twelve leafy vegetable species and the elements of CFSI in two Karen and two Lawa villages in northern Thailand. (AI = availability index; FUI = frequency of utilization index; PUI = plant parts used index; MFFI = multi-functional food use index; TSAI = taste score appreciation index; and FMRI = food-medicinal role index). The thicker the band, the more informants mentioned the relationship.

**Table 1 foods-09-01748-t001:** Twelve wild leafy vegetable species encountered in two Karen and two Lawa villages in northern Thailand, giving their scientific name, family, common name in English, ethnobotanical information, distribution, collection period, and their Cultural Food Significance Index (CFSI) values.

Scientific Name (Family); Voucher Number (Punchay # Deposited at CMUB)	Folk Name [29]	Traditional Use by the Karen (K) and the Lawa (L) [29]	Distribution	Collection Period	CFSI Score
*Oroxylum indicum* (L.) Kurz (Bignoniaceae) 566	Do ka (K) Dak ra wi (L)	Young shoots or fruits eaten raw or cooked. Inner bark grated and added to food for bitter taste (K, L).	Asia	All year	813
*Centella asiatica* (L.) Urb. (Apiaceae) 546	Sui po na do (K) Phak nhok (L)	Aerial parts eaten raw as a side dish for treating bruises (K, L).	Africa, Asia, North- and South America, and Oceania	All year	629
*Senegalia rugata* (Lam.) Britton and Rose (Leguminosae) 605	Pa chi (K) Kad ka ha (L)	Young shoots eaten raw or cooked with fish (K, L).	Asia and Oceania	All year	437
*Ficus auriculata* Lour. (Moraceae) 542	Ta kue po (K) Mae (L)	Shoots boiled as a side dish or cooked (K, L).	Asia	May–Oct	432
*Clerodendrum glandulosum* Lindl. (Lamiaceae) 304	Ko ko do (K) Tung lam (L)	Young leaves boiled and pressed to reduce bitter taste, then fried with eggs or cooked as soup (K, L).	Asia	Dec–Apr	392
*Spondias pinnata* (L. f.) Kurz (Anacardiaceae) 443	Pi sae (K) Kok (L)	Young leaves eaten raw as a side dish (K, L). Stems decocted for treating diarrhea (K).	Asia	Jun–Jan	379
*Lygodium flexuosum* (L.) Sw. (Lygodiaceae) 457	Ki ko do (K) Wu wia (L)	Shoots boiled as a side dish or cooked (K, L). Roots boiled as a beverage (tea) (K).	Asia and Oceania	May–Feb	378
*Oenanthe javanica* (Blume) DC. (Apiaceae) 564	Po a do (K) Tu klae (L)	Aerial parts eaten raw as a side dish (K, L).	Asia	Jun–Dec	365
*Acmella paniculata* (Wall. ex DC.) R.K.Jansen (Asteraceae) 594	Hor te mi (K) Tu plei (L)	Aerial parts eaten raw with chili paste or cooked (K, L). Roots chewed to treat toothache (K).	Asia	Jun–Jan	353
*Acmella uliginosa* (Sw.) Cass. (Asteraceae) 522	Hor te mi (K) Tu plei (L)	Aerial parts eaten raw with chili paste or cooked. Roots chewed to treat toothache (K, L).	Native pantropical	Jun–Jan	353
*Monochoria vaginalis* (Burm.f.) C.Presl (Pontederiaceae) 690	No do (K) Seuk lek (L)	Young shoots and petioles eaten raw or cooked (K, L).	Asia and North America (introduced)	May–Sep	336
*Musa acuminata* Colla (Musaceae) 536	Ya pa la (K) Lha wong pia (L)	Pseudostems chopped and cooked as soup/or used as fodder; inflorescences were eaten raw or cooked (K, L).	Throughout tropics in Africa, America, and Asia	All year	308

**Table 2 foods-09-01748-t002:** Proximate composition of the edible parts of selected wild food plant species (g/100 g fresh weight).

Species	Moisture Content	Ash	Protein	Fat	Fiber	Carbohydrate
*Acmella paniculata*	87.4 ± 1.6 ^c^	1.7 ± 0.2 ^bc^	2.92 ± 0.4 ^de^	0.55 ± 0.07 ^f^	2.17 ± 0.2 ^bc^	5.24 ± 0.8 ^c^
*Acmella uliginosa*	88.9 ± 1.5 ^c^	1.4 ± 0.2 ^abc^	3.58 ± 0.1 ^e^	0.34 ± 0.03 ^cd^	1.46 ± 0.3 ^ab^	4.17 ± 0.6 ^bc^
*Centella asiatica*	87.6 ± 1.7 ^c^	1.8 ± 0.2 ^bc^	2.71 ± 0.4 ^a^	0.32 ± 0.04 ^cd^	1.84 ± 0.3 ^abc^	5.69 ± 0.8 ^c^
*Clerodendum glandulosum*	76.3 ± 1.8 ^b^	1.8 ± 0.1 ^bc^	5.33 ± 0.4 ^f^	0.48 ± 0.04 ^ef^	3.01 ± 0.3 ^cd^	13.08 ± 1.0 ^e^
*Ficus auriculata*	86.6 ± 1.1 ^c^	1.9 ± 0.1 ^c^	2.89 ± 0.3 ^de^	0.34 ± 0.03 ^cd^	3.77 ± 0.3 ^e^	4.47 ± 0.4 ^bc^
*Lygodium flexuosum*	62.9 ± 2.6 ^a^	2.8 ± 0.2 ^d^	3.78 ± 0.3 ^e^	0.78 ± 0.07 ^g^	11.82 ± 0.7 ^e^	17.94 ± 1.3 ^f^
*Monochoria vaginalis*	94.8 ± 1.2 ^d^	1.1 ± 0.2 ^ab^	0.56 ± 0.1 ^ab^	0.11 ± 0.03 ^a^	0.79 ± 0.2 ^a^	2.63 ± 0.6 ^ab^
*Musa acuminata*	95.3 ± 1.0 ^d^	1.2 ± 0.2 ^ab^	0.48 ± 0.1 ^a^	0.10 ± 0.02 ^a^	1.36 ± 0.3 ^ab^	1.54 ± 0.4 ^a^
*Oenanthe javanica*	88.7 ± 3.0 ^c^	2.1 ± 0.5 ^cd^	1.85 ± 0.4 ^cd^	0.17 ± 0.05 ^ab^	2.87 ± 0.7 ^cd^	4.26 ± 1.3 ^bc^
*Oroxylum indicum*	86.8 ± 1.0 ^c^	0.8 ± 0.1 ^a^	1.46 ± 0.1 ^bc^	0.26 ± 0.02 ^bc^	2.03 ± 0.2 ^bc^	8.65 ± 0.7 ^d^
*Senegalia rugata*	85.7 ± 1.7 ^c^	0.8 ± 0.1 ^a^	4.77 ± 0.6 ^f^	0.41 ± 0.05 ^de^	3.98 ± 0.5 ^e^	4.32 ± 0.5 ^bc^
*Spondias pinnata*	75.6 ± 0.8 ^b^	4.0 ± 0.3 ^e^	3.61 ± 0.1 ^e^	0.74 ± 0.03 ^g^	3.84 ± 0.4 ^e^	11.93 ± 0.4 ^e^

^a–e^ In each column, different letters at the end of each value represent the differences of the mineral content among plant species (Tukey’s HSD test). The species sharing the same letter were not significantly different in their mineral contents.

**Table 3 foods-09-01748-t003:** Mineral composition (mg/100 g fresh weight) in twelve wild food plant species sampled in two Karen and two Lawa villages in northern Thailand. The figures are compared to the recommended dietary allowance (RDA) and tolerable upper intake level (UL) according to [59].

Species	Plant part	P	K	Ca	Mg	Na	Fe	Mn	Zn	Cu
*Acmella paniculata*	Aerial parts	40.5 ± 0.5 ^de^	356.2 ± 1.1 ^ab^	183.4 ± 4.5 ^c^	55.3 ± 2.0 ^ef^	1.8 ± 0.1 ^abc^	11.1 ± 1.2 ^e^	1.8 ± 1.1 ^ab^	6.7 ± 1.10 ^de^	0.04 ± 0.00 ^d^
*Acmella uliginosa*	Aerial parts	44.2 ± 0.6 ^f^	549.7 ± 9.9 ^cde^	147.8 ± 1.8 ^b^	56.5 ± 1.1 ^f^	0.7 ± 0.1 ^a^	10.7 ± 0.3 ^e^	7.1 ± 1.0 ^c^	1.9 ± 0.19 ^ab^	0.09 ± 0.00 ^f^
*Centella asiatica*	Leaves	32.2 ± 1.4 ^c^	377.0 ± 42.0 ^abc^	176.7 ± 10.5 ^bc^	104.4 ± 2.8 ^h^	21.2 ± 1.6 ^e^	4.1 ± 0.2 ^bc^	12.0 ± 0.7 ^de^	10.9 ± 1.86 ^f^	0.02 ± 0.00 ^b^
*Clerodendum glandulosum*	Young leaves	71.1 ± 2.9 ^h^	693.7 ± 11.9 ^e^	24.7 ± 3.1 ^a^	69.8 ± 3.2 ^g^	1.5 ± 0.1 ^abc^	2.2 ± 0.1 ^ab^	10.9 ± 1.5 ^d^	19.4 ± 2.63 ^h^	0.34 ± 0.01 ^g^
*Ficus auriculata*	Young leaves	48.6 ± 1.1 ^g^	447.2 ± 7.7 ^bcd^	338.1 ± 15.2 ^d^	50.1 ± 3.2 ^de^	4.8 ± 1.1 ^d^	5.5 ± 0.1 ^bc^	3.3 ± 0.3 ^b^	5.7 ± 0.34 ^cd^	0.05 ± 0.00 ^e^
*Lygodium flexuosum*	Young leaves	44.2 ± 1.9 ^f^	668.3 ± 21.8 ^e^	15.4 ± 0.8 ^a^	57.4 ± 3.6 ^f^	2.4 ± 0.4 ^c^	6.3 ± 4.3 ^cd^	14.0 ± 1.3 ^ef^	7.0 ± 0.68 ^de^	0.03 ± 0.00 ^cd^
*Monochoria vaginalis*	Leaves	16.4 ± 0.4 ^b^	253.4 ± 10.9 ^a^	28.5 ± 1.6 ^a^	12.4 ± 0.6 ^bc^	0.8 ± 0.1 ^ab^	9.3 ± 0.2 ^de^	15.7 ± 0.3 ^f^	3.0 ± 0.42 ^abc^	<0.01 ^a^
*Musa acuminata*	Psuedostem	9.3 ± 1.0 ^a^	591.0 ± 25.5 ^de^	2.6 ± 0.1 ^a^	8.0 ± 0.8 ^ab^	0.4 ± 0.1 ^a^	0.4 ± 0.1 ^a^	6.3 ± 0.9 ^c^	4.6 ± 0.5 ^bcd^	<0.01 ^a^
*Oenanthe javanica*	Aerial parts	43.5 ± 1.5 ^ef^	1291.3 ± 254 ^f^	170.2 ± 12.0 ^bc^	46.4 ± 0.8 ^d^	2.2 ± 0.1 ^bc^	7.0 ± 0.6 ^cd^	8.4 ± 0.5 ^c^	14.0 ± 0.5 ^g^	0.06 ± 0.00 ^e^
*Oroxylum indicum*	Fruits	31.0 ± 1.0 ^c^	235.3 ± 9.2 ^a^	3.0 ± 0.4 ^a^	14.2 ± 2.1 ^c^	0.5 ± 0.0 ^a^	0.3 ± 0.0 ^a^	0.2 ± 0.0 ^a^	6.4 ± 0.9 ^de^	0.41 ± 0.01 ^h^
*Senegalia rugata*	Young leaves	77.7 ± 1.5 ^i^	221.7 ± 11.2 ^a^	1.0 ± 0.1 ^a^	13.4 ± 1.0 ^bc^	0.9 ± 0.1 ^ab^	0.7 ± 0.0 ^a^	0.4 ± 0.0 ^a^	9.0 ± 1.1 ^ef^	0.01 ± 0.00 ^ab^
*Spondias pinnata*	Young leaves	37.8 ± 1.4 ^d^	244.8 ± 35.5 ^a^	982.9 ± 37.2 ^e^	6.0 ± 0.2 ^a^	1.2 ± 0.4 ^abc^	2.4 ± 0.4 ^ab^	8.1 ± 1.1 ^c^	1.4 ± 0.2 ^a^	0.02 ± 0.00 ^bc^
RDA ^1^ and UL ^2^										
Male		700‒(4000)	4700 *	1000‒(2500)	330‒(350) **	1500‒(2300)	8‒(45)	2.3‒(11)	11‒(40)	0.9‒(10)
Female		700‒(4000)	4700 *	1000‒(2500)	255‒(350) **	1500‒(2300)	18‒(45)	1.8‒(11)	8‒(40)	0.9‒(10)
Pregnancy		700‒(3500)	4700 *	1000‒(2500)	290‒(350) **	1500‒(2300)	27‒(45)	2.0‒(11)	11‒(40)	1‒(8)
Lactation		700‒(4000)	5100 *	1000‒(2500)	255‒(350) **	1500‒(2300)	9‒(45)	2.6‒(11)	12‒(40)	1.3‒(8)

^a–i^ In each column, different letters at the end of each value represent the differences in the mineral content among plant species (Tukey’s HSD test). The species sharing the same letter were not significantly different in their mineral contents. ^1^ Recommended dietary allowance in adults aged 19‒50 years (mg/day); ^2^ Tolerable upper intake level in adults aged 19‒50 years (mg/day) [59]. * For the absence of a UL, extra caution may be warranted in consuming levels above the recommended intake. ** The UL for magnesium represents intake from pharmacological agents only and does not include intake from food and water.

**Table 4 foods-09-01748-t004:** Percentages of contribution to the recommended dietary allowance (RDA) (mg/day) [59], for adults, of macrominerals (P, K, Ca, Mg, and Na) and trace minerals (Fe, Mn, Zn, and Cu).

**Scientific Name**	**% of RDA (Male/Female)**				
	**P**	**K**	**Ca**	**Mg**	**Na**
*Acmella paniculata*	5	7	18	16/21	0.12
*Acmella uliginosa*	6	11	14	17/22	0.04
*Centella asiatica*	4	8	17	31/41	1.41
*Clerodendum glandulosum*	10	14	2	21/27	0.10
*Ficus auriculata*	7	9	33	15/19	0.32
*Lygodium flexuosum*	6	14	1	17/22	0.16
*Monochoria vaginalis*	2	5	2	3/4	0.05
*Musa acuminata*	1	12	0.2	2/3	0.02
*Oenanthe javanica*	6	27	17	14/18	0.14
*Oroxylum indicum*	4	5	0.3	4/5	0.03
*Senegalia rugata*	11	4	0.1	4/5	0.06
*Spondias pinnata*	5	5	98	1/2	0.08
**Scientific Name**		**% of RDA (Male/Female)**			
		**Fe**	**Mn**	**Zn**	**Cu**
*Acmella paniculata*		138/61	78/100	60/83	4
*Acmella uliginosa*		133/59	308/394	17/23	10
*Centella asiatica*		51/22	521/666	99/136	2.2
*Clerodendum glandulosum*		27/12	389/605	176/242	37
*Ficus auriculata*		68/30	143/183	51/71	5
*Lygodium flexuosum*		78/35	608/777	63/87	3
*Monochoria vaginalis*		116/51	682/872	27/37	<1
*Musa acuminata*		5/2	273/350	41/57	<1
*Oenanthe javanica*		87/38	365/466	127/175	6
*Oroxylum indicum*		3/1	8/11	58/80	45
*Senegalia rugata*		8./3	17/22	81/112	1
*Spondias pinnata*		30/13	352/450	12/17	2

The percentages of RDA values for macrominerals and trace minerals are compared with the consumptions of 100 g of fresh leafy vegetables. The percentages of RDA values of P, K, Ca, Na, and Cu, in male and female adults (19–50 years) are equal.

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
