# Peer review of "Nutrient and Mineral Compositions of Wild Leafy Vegetables of the Karen and Lawa Communities in Thailand"

_foods, 2020, doi:10.3390/foods9121748_

Round 1

Reviewer 1 Report

Comments, review – Nutrient and mineral compositions of wild leafy vegetables of the Karen and Lawa in Thailand

Dear authors

An interesting and important manuscript on the macro- and mineral composition of leafy vegetables; I have a few comments and suggestions for improvement; in particular, I would like to see the mineral concentrations per (estimated) fresh weight and their comparisons with RDI values.

Abstract: you write ‘Contents of potassium, calcium, zinc, and iron in several of the wild leafy vegetables were high enough to cover the daily recommended dietary intakes of adults’ – however this depends on how much the Karen and Lawa eat of those wild leafy vegetables (WLV). In other words, how much (in the mean) do one need to eat of a leafy vegetable (LV) in order to reach the daily recommendation of iron?  And I guess they then don’t eat the vegetables in dried form; therefore, please calculate the values (Table 3) back as mg per fresh weight – an additional table 4 showing the values of table 3, but per wet weight.

Methods: Line 109_ rinsed and cleaned (cleansed)

Line 110: how did you dry your sample? And how did you calculate the moisture of your samples? The exact information (in gram) of moisture lost would allow you to calculate the analysed nutrients per ‘estimated’ fresh weight.  It’s here not clear why you write ‘samples were dried’, and later you analysed ‘fresh’ samples for moisture, ash, protein, fat,…and the ash for minerals; for what do you need the dried samples? Or do you mean that you just dried them from the deionised water used for cleaning - please clarify

Line 113: ‘were analysed raw’ - you mean the dried or un-dried samples? Please clarify

Line 130: please provide the information (company, model) of the AAS.

Results: Table 1: Very little was said about Table 1 in the results; is there nothing more to say?

Figure 1: nice picture, but very difficult to follow the results in detail (e.g. %); the question remains unanswered how often and, more importantly, how much of these twelve leafy vegetables (LV) the Karen usually eat (consume).

Figure 2: very confusing; how can you have two scales, moisture in g/ 100 g and everything else ? in g /100 g dry weight; if you give the values of protein, fat and fibre per 100 g fresh weight as well as the moisture, you will of course get smaller, but instead the actual concentration in the leafy vegetables; alternatively you could also show the values in a table.  And for the calculation of the mineral concentration per wet weight, you need the exact moisture content for the (re)calculation anyway

Line 160: yes the exact values would be necessary to calculate concentrations per fresh weight

Line 162…: all the other values are per dry weight: e.g. The ash contents, which indicated the mineral content, ranged from 162 0.79–4.28 g/100g ?

I don’t mind, and would urgently suggest to present both, values per dry AND (more important) per fresh weight

Line 180: The concentrations of macro-minerals (P, K, Mg, Ca, Na) and micro-minerals (Fe, Cu, Zn, Mn) - you mean minerals that occur in larger quantities vs. those that occur in traces - very unusual term ‘macro- and micro minerals’.

Table 3: I would urgently suggest to calculate all the mineral contents per ‘fresh weight’ ; as you have analysed the moisture content (moisture lost with drying), this can be easily done; an additional table (table 4) with data on minerals (mg/ estimated fresh) allows you a real comparison of mineral content in edible vegetables with the recommend daily intakes; and here I would suggest to use the FAO tables for the RDI as reference.

WHO/FAO. Vitamin and Mineral Requirements in Human Nutrition: Report of a Joint FAO/WHO Expert Consultation, Bangkok, Thailand, 21–30 September 1998; WHO: Geneva, Switzerland, 2004.

At the moment the table 3 is only meaningful for the comparison of minerals between VL species (in dried form); theoretical recommendations and assessments regarding the recommended daily intake of minerals are only possible with values related to the fresh weight. 

Nobody (including the Karen) will eat the vegetables as dried food; therefore, a comparison of (‘only’) mg mineral per gram dried weight makes not much sense to compare with the RDI – the dried vegetables are much higher concentrated in minerals. Please rRead the following study published in Foods (2019) as an example how to present values per fresh weight:

Gowele, V. F., Kinabo, J., Jumbe, T., Kirschmann, C., Frank, J., & Stuetz, W. (2019). Provitamin A Carotenoids, Tocopherols, Ascorbic Acid and Minerals in Indigenous Leafy Vegetables from Tanzania. Foods, 8(1), 35.

Line 180-194: The additional conversion and presentation of the minerals by fresh weight could give a different picture according to the different moisture content; lygodium flexuosum for example has a relatively low moisture content;

After all, you do not compare the actual mineral concentrations with those of RDA and UL; you only represent the RDA and UL values.

Line 194: 3.31mg not mh; however, what do all these values have to do with nutrition or assessment in relation to the RDA; I would like to know how much of each LV would have to eat to achieve the RDA; or, how much of the RDA (in %) would be achieved with 100 grams of fresh (fresh weight) leafy vegetables?

Discussion

Line 204-44: Under 4.1 and 4.2 they unfortunately do not say how much of the wild leafy vegetables are actually eaten, nor do they give any information about the habits of further processing, e.g. to what extent the LV are dried and stored; or are the LV mainly consumed fresh by the Karen and Lawa? Are LVs really consumed as vegetables or just added to food? What is the real proportion of LV in the Karen diet?

4.3. Line 245-73

You can hardly say anything about this: the evaluation of the mineral content per dry weight compared to the recommended daily intake, without giving an idea of how much of it is eaten, makes no sense; or do you believe or assume that the Karen or Lawa eat an average of 100 grams of dried LV daily? In other words, how much of the dried vegetables would you need to eat each day to reach the RDA of the reported minerals?

Iron, zinc, calcium and magnesium are particularly important in human nutrition; iron and zinc deficiency are frequently reported among ethnic minorities and migrants (e.g. Karen) in South East Asia, including Thailand. 

Also take into account that you may also compare the mineral values with those of other publications, which are given per 100 gram wet weight (ww), for example [14]:

García-Herrera, P.; Sánchez-Mata, M.C.; Cámara, M.; Fernández-Ruiz, V.; Díez-Marqués, C.; Molina, 330 M.; Tardío, J. Nutrient composition of six wild edible Mediterranean Asteraceae plants of dietary 331 interest. J. Food Compos. Anal. 2014, 34, 163‒170       

Line 266: Manganese: For me it is difficult to think of manganese poisoning through the consumption of LV. Foods rich in manganese are (whole grain) cereal products such as bread, wheat germ, oatmeal, millet or rice, but also pulses, linseed and nuts as well as green leafy vegetables and dark berries (blueberries, aronia berries) or prunes. Although black tea contains a lot of manganese (690 µg/100 ml), it is not readily bioavailable due to its high tannin content. Coffee (drink) is also a source of manganese (80 µg/100 ml) in relation to the usual drinking quantities.

Conclusion:

Line 278: ‘Some the studied wild plant species accumulated mineral from the environment to levels that exceed the daily tolerable intake rate, producing health-related risks - here you can't really say anything concrete about it, because you haven't calculated for any single plant the amount of mineral for e.g. the consumption of 100 grams fresh leafy vegetables.

The presentation of minerals per dry weight is fine, but for the evaluation of consumption, values per fresh weight should be described and discussed in relation to the RDA; or, should at least be described how much of the dried LV is actually consumed.

Reviewer 2 Report

Dear Editor!

 I checked the manuscript entitled ‘Nutrient and mineral compositions of wild leafy vegetables of the Karen and Lawa in Thailand’ by Kittiyut Punchay, Angkhana Inta, Pimonrat Tiansawat, Henrik Balslev and Prasit Wangpakapattanawong'. The manuscript contains reasonably good and interesting results. Introduction part is focused on the paper, methodologies are given in detail. Discussion chapter needs some further elaboration.

Authors should consider comments below, among others, when they are improving the manuscript.

Abstract:

Now the abstract seems too general, I suggest authors include the most important findings to make the abstract more informative.

Introduction:

I suggest that authors improve the aim of the study with more detalied description, focusing on research work done.

Material and methods:

I suggest to move Table 1 to the beginning of Materials and methods chapter. Now the Material and methods section lacks the most important information regarding the studied plants.

Lines 108-109:

The sentence is not clear, I suggest to improve it: The homogenoius samples  were susampled. 

Line 143: Turkey's test – correct to Tukey's test

Results: I suggest to replace concentration with contents which is appropriate expression i.e. a concentration is an amount of any type per volume of liquid or gas system, whereas content is an amount of any type per mass of liquid or gas or solid system.

Discussion:

I suggest authors make some comparisons of composition of their own results regarding wild leafy vegetables to the composition of for example leafy vegetables like spinach and  lettuce and on the other hand asparagus for shoot type vegetable.

Conclusions:

I suggest to rewrite it, emphasize the most important findings in terms of which constituents are the most important form nutritional point of view.

Reviewer 3 Report

The structure of the manuscript is logical and clear. The need for the study is justified and the conclusions are well formulated. The manuscript is suitable for publication in the journal "Foods", but however, some corrections would need to be made before final publication.

  • Review the numbering of the chapters (eg: “2.1 Selection of wild leafy vegetable species” and “2.1 Study sites” have the same number).
  • In the "Study sites" chapter, you should mention the natural conditions in one or two sentences. For example, is it a tropical rainforest, mountains, etc. In the tropics, there are no interruptions to the plants during the growing season and green plants are available at any time. Did this also apply to these 12 species? In other words, could fresh leaves or young shoots be obtained from these 12 plant species throughout the year, or were there interruptions in obtaining them in a few months of the year?
  • Pieroni (2001) uses the term “Utilization Frequency Index (UFI)” and you use “frequency of use (FUI) refering to Pieroni (2001). Although there is essentially no difference in the concept, you should then explain why you do not use "UFI"?
  • In terms of methodology, you refer to your previously published articles. However, you should even now also say, which month the fieldwork took place and in which month did you pick these plant samples?
  • Under the "Preparation of samples", describe how fresh plant samples were stored from collection to transport to the laboratory. And how long they were stored before being taken to the lab. The laboratory has evaluated carbohydrates, proteins and moisture content - could storage have affected these parameters in the meantime?
  • Table 3 should be harmonized. It is currently set to "," to denote a thousand. I would recommend removing these commas as has been done for the Ficus auriculata “K” number (see 3312 ± 57de). The reason is that it misleads the viewer if there are both commas and dots in one column (see the “Ca” column) and therefore I recommend removing the commas altogether and leaving only the “.” when is numbers after the decimal point.
  • In Table 1, the English “Common name” adds nothing. Rather, there must be a "folk name" of the plant - the name that local people said.
  • In this sentence, the reference to waste burning is erroneous and should be removed (line 273): “The high concentration of zinc in wild plants may come from zinc-rich soils, or burning of waste [69]”. Looking at the original source ([69] Abbasi, A.M.; Shah, M.H.; Khan, M.A. Wild Edible Vegetables of Lesser Himalayas: Ethnobotanical and Nutraceutical Aspects; Springer International Publishing: Cham, 2014.) then burning of waste is not mentioned (page 47): “Water may be polluted by Zn due to the presence of its large quantities in wastewater of industrial plants. One of the consequences is that rivers are depositing zinc-polluted sludge on their banks. Plants often have a zinc uptake that their systems cannot handle, due to which on zinc-rich soils only a limited number of plants have a chance of survival. It can also interrupt the activity in soils, as it negatively influences the activity of microorganisms and earthworms (Simeonov and Sargsyan 2008).”.
  • I understand that you based your study primarily on medical ethics (line 100): “International Guidelines for Human Research Protection” (“World Medical Association. (2001). World Medical Association Declaration of Helsinki. Ethical principles for medical research involving human subjects. Bulletin of the World Health Organization, 79(4), 373–374.”). However, as these are indigenous peoples, you should also follow the Declarations adopted to protect them. So, if you followed these principles, you could also say that and refer to it (eg suitable for reference: ”International Society of Ethnobiology (2006). International Society of Ethnobiology Code of Ethics (with 2008 additions). http://ethnobiology.net/code-of-ethics/ “)

Round 2

Reviewer 1 Report

Dear authors: thanks for the edited manuscript, well done; only one last comment:

you present now all values od minerals per 100 gram fresh weight, well done; and you also compare the concentrations measured with the RDA -Table 4: Percentage of contribution to RDA...for adults; however,  you forgot to mention that you compared the consumption of 100 grams of fresh leafy vegetables with RDA values for macrominerals and trace minerals -  I think it should be clear that you assume the amount of 100 grams of fresh wild leafy vegetables; please add (complete) this information in the title and/or in the footnotes of table 4.

Author Response

Dear Reviewer 1

According to the revision, we added these sentences in the footnote of Table 4 (new line 310-312) as follows:

"The percentages of RDA values for macrominerals and trace minerals are compared with the consumptions of 100 grams of fresh leafy vegetables. The percentages of RDA values of P, K, Ca, Na, and Cu, in male and female adults (19–50 years) are equal."

Thank you for comments that helped us improve the quality of this manuscript.

Kittiyut Punchay